# International Travel as a Risk Factor for Carriage of Extended-Spectrum β-Lactamase-Producing *Escherichia coli* in a Large Sample of European Individuals—The AWARE Study

**DOI:** 10.3390/ijerph19084758

**Published:** 2022-04-14

**Authors:** Daloha Rodríguez-Molina, Fanny Berglund, Hetty Blaak, Carl-Fredrik Flach, Merel Kemper, Luminita Marutescu, Gratiela Pircalabioru Gradisteanu, Marcela Popa, Beate Spießberger, Laura Wengenroth, Mariana Carmen Chifiriuc, D. G. Joakim Larsson, Dennis Nowak, Katja Radon, Ana Maria de Roda Husman, Andreas Wieser, Heike Schmitt

**Affiliations:** 1Institute and Clinic for Occupational, Social and Environmental Medicine, University Hospital, LMU Munich, 80336 Munich, Germany; laura.wengenroth@med.uni-muenchen.de (L.W.); dennis.nowak@med.uni-muenchen.de (D.N.); katja.radon@med.uni-muenchen.de (K.R.); 2Institute for Medical Information Processing, Biometry and Epidemiology—IBE, LMU Munich, 81377 Munich, Germany; 3Pettenkofer School of Public Health, 81377 Munich, Germany; 4Department of Infectious Diseases, Institute of Biomedicine, The Sahlgrenska Academy, University of Gothenburg, 40530 Gothenburg, Sweden; fanny.berglund@gu.se (F.B.); carl-fredrik.flach@microbio.gu.se (C.-F.F.); joakim.larsson@fysiologi.gu.se (D.G.J.L.); 5Centre for Antibiotic Resistance Research (CARe), University of Gothenburg, 40530 Gothenburg, Sweden; 6Centre of Infectious Disease Control, National Institute for Public Health and the Environment, 3721 MA Bilthoven, The Netherlands; hetty.blaak@rivm.nl (H.B.); merel.kemper@rivm.nl (M.K.); ana.maria.de.roda.husman@rivm.nl (A.M.d.R.H.); heike.schmitt@rivm.nl (H.S.); 7Department of Microbiology and Immunology, Faculty of Biology, University of Bucharest and the Academy of Romanian Scientists, 050657 Bucharest, Romania; luminita.marutescu@bio.unibuc.ro (L.M.); gratiela.gradisteanu@icub.unibuc.ro (G.P.G.); marcela.popa@bio.unibuc.ro (M.P.); carmen.chifiriuc@bio.unibuc.ro (M.C.C.); 8Earth, Environmental and Life Sciences Section, Research Institute of the University of Bucharest, University of Bucharest, 030018 Bucharest, Romania; 9German Centre for Infection Research (DZIF), Partner Site Munich, 80336 Munich, Germany; spiessberger@mvp.uni-muenchen.de (B.S.); wieser@mvp.uni-muenchen.de (A.W.); 10Max von Pettenkofer Institute, Faculty of Medicine, LMU Munich, 81377 Munich, Germany; 11Department of Infectious Diseases and Tropical Medicine, LMU University Hospital Munich, 80802 Munich, Germany; 12Comprehensive Pneumology Center Munich (CPC-M), German Center for Lung Research (DZL), 80336 Munich, Germany

**Keywords:** antibiotic resistance, antimicrobial resistance, risk factors, ESBL *E. coli*, travels

## Abstract

Antibiotic resistance (AR) is currently a major threat to global health, calling for a One Health approach to be properly understood, monitored, tackled, and managed. Potential risk factors for AR are often studied in specific high-risk populations, but are still poorly understood in the general population. Our aim was to explore, describe, and characterize potential risk factors for carriage of Extended-Spectrum Beta-Lactamase-resistant *Escherichia coli* (ESBL-EC) in a large sample of European individuals aged between 16 and 67 years recruited from the general population in Southern Germany, the Netherlands, and Romania. Questionnaire and stool sample collection for this cross-sectional study took place from September 2018 to March 2020. Selected cultures of participants’ stool samples were analyzed for detection of ESBL-EC. A total of 1183 participants were included in the analyses: 333 from Germany, 689 from the Netherlands, and 161 from Romania. Travels to Northern Africa (adjusted Odds Ratio, aOR 4.03, 95% Confidence Interval, CI 1.67–9.68), Sub-Saharan Africa (aOR 4.60, 95% CI 1.60–13.26), and Asia (aOR 4.08, 95% CI 1.97–8.43) were identified as independent risk factors for carriage of ESBL-EC. Therefore, travel to these regions should continue to be routinely asked about by clinical practitioners as possible risk factors when considering antibiotic therapy.

## 1. Introduction

Extended-spectrum β-lactamases (ESBLs) are plasmid-mediated enzymes that inactivate β-lactam antibiotics, posing a significant therapeutic challenge in the treatment of both hospital and community-acquired infections [1]. Infections with ESBL-producing *E. coli* (ESBL-EC) often require therapy with last-resort antibiotics, increasing both the risk of resistance and the associated healthcare costs [2,3]. Resistance to last resort antibiotics further limits treatment options and is associated with prolonged hospital stays and increased mortality [4]. An increase in the prevalence of ESBL-EC, in both community and healthcare settings, is now observed worldwide: the current global prevalence of healthy individuals with ESBL-EC from 2003 to 2018 is estimated to be 16.5%; having increased from 2.6% in 2003–2005 to 21.1% in 2015–2018 [5]. In 2019, we estimated the prevalence of these bacteria in the general population of three European countries, and we found it to be 13% in Romania, 8% in Germany, and 6% in the Netherlands [6]. For comparison, the current prevalence in Europe is 6% [5].

The development and spread of antibiotic resistance (AR) is correlated with the use of antibiotics in the healthcare sector and in the agriculture and husbandry sectors [1,3,7,8]. Antibiotic therapy is also a risk factor for carriage of AR by individuals. Other potential risk factors include: travels to high-risk areas for AR [2,9,10,11,12,13,14,15,16,17,18,19,20,21,22,23,24,25,26,27,28], consumption of food contaminated with AR bacteria [29,30], a poorer health status that leads individuals into being treated with antibiotics or at healthcare facilities increasing their exposure to AR bacteria [23,26], and occupation where the workplace might potentially increase exposure to antibiotics or AR bacteria, such as working at animal markets, dairy facilities, farms, slaughterhouses, wastewater treatment plants, and healthcare facilities [31,32,33,34,35,36,37,38,39,40,41,42,43,44,45,46]. However, most of the studies examining potential risk factors focus on high-risk populations, such as travelers [10,12,16,20,22,26,27,47], healthcare workers and patients [40,45,46,48,49,50], swimmers [51,52,53], farmers [33,34,35,36,38,39,41,43,44], and slaughterhouse workers [32], and often use small, convenient samples of, e.g., students [2,18,19,23]. However, risk factors for AR in the general population have not yet been sufficiently investigated. This is of great importance for developing preventive measures and antibiotic therapy policies.

As part of the larger AWARE study [6,54], this study aimed to explore, describe, and characterize potential risk factors for carriage of ESBL-EC in a large sample of European individuals recruited from the general population in three countries with a different prevalence of AR, i.e., Germany, the Netherlands, and Romania.

## 2. Materials and Methods

### 2.1. Study Design and Participants

The study population comes from participants enrolled in the large trans-European cross-sectional AWARE study (Antibiotic Resistance in Wastewater: Transmission Risks for Employees and Residents around Wastewater Treatment Plants). The full methodology of this project has been previously described [6,54]. The subset of the data used in these analyses corresponds to individuals from the general population living more than 1000 m away from a local WWTP, and, thus, not exposed to potential AR bacteria coming from such facilities. Data collection took place from September 2018 to March 2020 in Southern Germany, the Netherlands, and Romania. Having age between 16 and 67 years was an inclusion criterion.

In Southern Germany, we recruited participants using households as the unit of recruitment. We obtained household participant information from local civil registries. Invitation letters were mailed to all individuals older than 16 years of age within the household. For locations where we could not obtain participant information through the civil registries, invitation letters were dropped in household mailboxes by members of the study team. Aids in recruitment included two reminder letters, articles about the project in the local newspaper, recruitment campaigns via Facebook, and a raffle of shopping vouchers worth EUR 1500 in total for participants who completed the study. In the Netherlands, the offices of general practitioners served as recruitment points. We used ArcGis™ [55] to identify all postal addresses in a 500-m radius from 22 different General Practitioners’ (GP) practices and then we randomly retrieved contact information for 200–500 households per GP practice using the Dutch Personal Records Database. The invitation to participate was addressed to all members of these households aged over the age of 16 (conform the conditions for General Data Protection Regulation data use). All participants who completed the study received a shopping voucher worth EUR 20. In Romania, we identified participant households and invited participants through door-to-door visits. 

Ethics approval was obtained from the Ethics Committee of the University of Munich (LMU) (Project-No. 17-734) and the Research Ethics Committee of the University of Bucharest (Registration-No. 164/05.12.2017). The ethics board in the Netherlands exempted this study for ethical approval under the Dutch Medical Research Involving Human Subjects Act (WMO; Committee: Medisch Ethische Toetsingscommissie, number of confirmation: 19-001/C). All subjects gave their informed consent for inclusion before they participated in the study. The study was conducted in accordance with the Declaration of Helsinki of 1975, revised in 2013.

### 2.2. Variables of Interest

#### Potential Risk Factors

Participants were asked to complete an online questionnaire [54] containing questions about socio-demographic characteristics, including date of birth (used to operationalize age in years), sex (female, male), educational level (according to the national educational system), and country of residence (Germany, the Netherlands, or Romania). The questionnaire also included questions about potential risk factors for carriage of ESBL-EC in the past year, such as: job history; hospital and farm visits (no, yes); contact with animals (no, yes); contact with patients or human tissues at work (never, rarely, sometimes, often, always); use of antibiotics and antacids (no, yes, do not know); self-reported health status (poor, fair, good, very good, excellent); self-reported frequency of diarrhea (never, rarely, sometimes, often, always); surgeries (no, yes); and international travel to Europe, Asia, North Africa, Sub-Saharan Africa, North America, Central America or Mexico, South America, and Australia or Oceania (never, once, 2–3 times, more than 3 times, do not know). The details on how these variables were chosen have been previously published [54].

Educational level was explored using national educational system levels and then dichotomized into low (pre-primary education to lower secondary education) or high educational level (upper secondary education to Doctoral or equivalent) according to the Standard Classification of Education (ISCED) [56,57,58]. Variables using a frequency scale with five levels were reduced to two levels in the case of frequency of diarrhea (never, rarely, or sometimes/often or always) and of self-reported health status (good, very good or excellent/fair or poor), and in the case of patient contact and of work with human tissues into three levels (never/rarely or sometimes/often or always). In questions including a “do not know” option (antibiotics and antacid intake, travels to Europe), this option was coded into the “no” category considering that the proportion of participants choosing this option was very low (3.1% for antibiotic intake, 2.9% for antacid intake, 0.1% for travels to Europe). We show descriptive counts for international travel variables as we collected the questionnaire data, i.e., using the following frequency scale for travel in the past 12 months: “never”, “once”, “2–3 times”, “more than 3 times”, “do not know”. For inferential analysis using regression models, these variables were collapsed into two levels: “never” and “at least once”. For the regression models, travels to Central and South America were collapsed into one variable. Additionally, we constructed a travel score considering travel to Asia, North Africa, Sub-Saharan Africa, Central America or Mexico, South America, and the European countries Italy, Bulgaria, Greece, and Slovenia as high-risk areas for AR. The travel score adds one point for travelling once, two points for travelling 2–3 times, and 3 points for travelling more than 3 times to any of these areas in the past year, while “never” was translated into zero points.

### 2.3. Outcome of Interest

In the Netherlands, all recruited participants were asked to provide a stool sample using a stool sample kit. In Germany and Romania, only participants who completed the online questionnaire were asked to provide a stool sample. After sampling, stool samples were kept refrigerated, transported in cooling boxes (2 °C to 8 °C), and processed within 24 h. Samples were inoculated directly into TBX (only in the Netherlands and Romania) or MacConkey (in Germany) agar plates (for *E. coli)*, and on ChromID^®^ ESBL (for ESBL-EC) and incubated at 36 °C ± 1 °C for 24–48 h. In case of positive results for ESBL-EC, 2 separate isolates per sample were collected from the selective ESBL plate for antibiotic resistance phenotype confirmation, and identification using MALDI-TOF MS (Matrix Assisted Laser Desorption Ionization-Time of Flight Mass Spectrometry). ESBL confirmatory tests were performed using cefotaxime and ceftazidime disks, alone and combined with clavulanate, following guidelines from the Clinical Laboratory Standards Institute (CLSI) [59]. The test was considered positive for strains showing a 5 mm increase in zone diameter in the presence of clavulanate. Stool sample results were coded binarily as positive or negative and included in the analyses.

### 2.4. Statistical Analyses

We used a Mann–Whitney test for observing differences in non-normally distributed numerical variables (age and travel score) and the Fisher’s exact test for differences in proportions (all the other variables). Variable selection was performed using a combination of bivariate analysis results (*p*-value ≤ 0.2) and expert opinion. We regressed carriage of ESBL-EC on a set of potential risk factors using two logistic regression models. The first model included sociodemographic variables (age, sex, educational level, and country of residence), frequency of diarrhea, antibiotics use, and travel score. The second model was similar to the first one, except that, instead of the travel score, it included each geographical area as we assessed them in the questionnaire, with “Central America or Mexico” and “South America” collapsed into one variable. We report both crude and adjusted estimates for both models. Missing values were handled by multiple imputation where the missing mechanism was missing at random (MAR) or missing completely at random (MCAR). MAR means that the probability of the data being missing is not due to unobserved data, conditional on the data that were collected. MAR is the second-best scenario for multiple imputation after MCAR, which occurs when the probability of the data being missing does not depend on the observed or unobserved data, and is, thus, the best scenario for multiple imputation [60]. Multiple imputation diagnostic tables can be found in the Appendix A. Inverse probability of sample weights was used to adjust for non-response by country [61,62]. We present model results in odds ratios (OR) with the corresponding 95% confidence intervals (CI). All analyses were performed in R version 4.1.0 [63].

## 3. Results

### 3.1. Study Population

In Germany, we invited 3153 residents (response 11%), while in the Netherlands we contacted 13,918 identified individuals by postal service, of which 10,448 were eligible by age (response 6%), and in Romania we invited 280 residents (response 54%). A total of 1183 participants were included in the analyses: 333 from Germany, 689 from the Netherlands, and 161 from Romania. The average prevalence of ESBL-EC carriage across the three countries was 7.5%, which corresponds to 8.4% in Germany, 6.1% in the Netherlands, and 12.6% in Romania. A total of 109 participants (95 in Germany, 3 in the Netherlands, and 11 in Romania) did not hand in a stool sample or had non-valid stool samples (9.2%). The large proportion of missing stool samples in Germany stems from having a short window for sample collection and transportation in this location, with which many participants failed to hand in the sample. This, however, did not happen in the Netherlands or Romania where samples were to be brought to GP practices within a 500-m distance from people’s homes collected by door-to-door visits.

The majority of participants in the overall sample were women (59.4%), middle-aged (median age 48 years, IQR 35–59), and highly educated (66.5%). Most participants reported no major risk factors for AR in the past year: no hospital visits neither as patient (92.9%), nor as professional (96.5%) or visitor (97.9%), no patient contact (73.6%), no use of antibiotics (76.1%) or antacids (77.2%), no surgeries (95.5%), no or infrequent diarrhea (94.2%), no work with human tissues (75.4%), no work with animals (96.5%), no work at a farm (99.0%), no work at a slaughterhouse (99.8%), no work with manure (97.0%), no farm visits (89.3%), and no animal contact (has no horses: 97.0%, has no dogs: 77.2%, has no cats: 75.7%). Additionally, most participants reported a health status from good to excellent (86.5%). Although a little more than two thirds of the study population reported travelling within Europe at least once in the past year (71.7%), they rarely traveled outside of the European continent: Australia or Oceania (1.0%), Central America (2.0%), South America (1.9%), Sub-Saharan Africa (2.4%), North America (3.6%), Northern Africa (4.2%), or Asia (7.2%). The proportion of population characteristics for individuals with a positive stool sample for ESBL-EC were similar as for the whole study population (Table 1 and Table 2). 

### 3.2. Risk Factors for ESBL-EC Carriage

Descriptive analyses including data from all study centers showed that ESBL-EC positive participants had higher education and were less likely to have a dog as a pet (Table 1). Furthermore, they were more likely to have had traveled at least once in the past year to Sub-Saharan Africa, Northern Africa, Asia, or North America according to bivariate analyses.

Country-specific analyses showed that travels to Northern Africa were associated with ESBL-EC carriage in the German sub-population, while an association was identified in the Dutch sub-population for traveling to Northern Africa, Sub-Saharan Africa, or Asia. In the Romanian subpopulation, high educational level, not having a dog as a pet, and working with human tissues were factors associated with ESBL-EC carriage. The travel score for travel to geographical areas with a known high-risk for AR, was significantly higher in the overall and Dutch ESBL-EC positive populations (*p*-value 0.02 and 0.001, respectively), compared to participants without ESBL-EC carriage (Table 2). 

Confirming descriptive and bivariate results, self-reported travel to North Africa, Sub-Saharan Africa, and Asia at least once in the past year were identified as independent risk factors for ESBL-EC carriage in our study population, both in crude and adjusted models (Figure 1). A summary of the adjusted estimates for travel to different geographical areas can be seen in Figure 2. 

On average, participants were about four times more likely to be carriers of ESBL-EC after travelling at least once in the past year to Northern Africa (adjusted OR 4.03, 95% CI 1.67–9.68), Sub-Saharan Africa (adjusted OR 4.60, 95% CI 1.60–13.26), and Asia (adjusted OR 4.08, 95% CI 1.97–8.43, Appendix A), compared with no travels to these regions. Although participants were twice as likely to be ESBL-EC carriers after traveling to North America, we could only identify a statistically significant association in the crude model (OR crude 2.79, 95% CI 1.17–6.67 vs. OR adjusted 2.40, 95% CI 0.94–6.09). The model including the travel score confirms these findings (Figure 1, Appendix A): Participants were 28% more likely to be ESBL-EC carriers when their travel score increased by one point, i.e., when they traveled at least once to any of the pre-specified high-risk areas for AR (adjusted OR 1.28, 95% CI 1.01–1.64, Appendix A).

## 4. Discussion

In this study, we found that destination for travels made during the past year is an important personal risk factor for carriage of ESBL-EC in the general population, especially North Africa, Sub-Saharan Africa, Asia, and—to some extent—North America. Other studies in risk populations have found similar results: some of these studies indicate that the prevalence of ESBL-EC acquisition is worryingly high in visitors returning from India, China and Southeast Asia, Middle East, Northern Africa, and Central and South America [64,65]. For European residents, travel outside of Europe was identified as a major travel risk factor [17]. A 2017 prospective study performed on Dutch travelers (n = 2001) found out that 34.3% of participants who were ESBL negative before travel, became positive for ESBL-EC during their travels, with the highest number being among participants travelling to Southern Asia [13]. 

We also found some differences in the country-specific travel patterns. By having collected a large sample size in The Netherlands, we were able to identify that this sub-population is at higher risk of ESBL-EC carriage when travelling to North Africa, Sub-Saharan Africa, and Asia within the past year. These results are comparable to those of a recently published large cross-sectional study of the Dutch general population, which identified traveling to Africa and Asia as independent risk factors for ESBL-EC carriage [66]. We found similar patterns in our German study population, where participants are at higher risk of ESBL-EC carriage after travels to Northern Africa and North America within the past year. Given that the national estimated prevalence of ESBL-EC causing urinary tract infections in the U.S. is 15.7%, ranging from 10.6% in the West North Central states to as high as 29.6% in the Mid-Atlantic states [67], our finding that travelers to North America were also at increased risk is not surprising. Conversely, in Romania, although the prevalence is already high, we found that the travel frequency is lower, therefore limiting our ability to analyze the effect of travel on ESBL-EC carriage in this subpopulation. Most of the Romanian participants reported not having travelled internationally at all within the past year. These findings suggest that the role of travel is country or context dependent.

The sewage surveillance data regarding the AR are in line with the estimated global burden of this threat. Current estimates indicate that the presence of AR genes found in the sewage is alarmingly at the highest level in Africa followed by Asia [68]. Models from sewage surveillance data show that the predicted clinical resistance to aminopenicillin, fluoroquinolones, and third generation cephalosporins are also at the highest resistance levels in Africa, followed by Asia [69]. These results from sewage surveillance data are in line with estimated global burden of disease from AR. The percentage of resistant isolates and the estimated death rate from AR *E. coli* have been reported to be at the highest in South Asia, followed by Sub-Saharan Africa [70]. Even though there have been some efforts in starting and maintaining clinical and sewage surveillance of AR bacteria in some countries of Africa and Asia [71], data on AR in these areas are still lacking to a large extent [70]. Some of these efforts include stewardship and surveillance programs in Ethiopia [72] and Ghana [73], or more generally in the African [74,75] and Asian regions [76,77,78]. The World Health Organization Global Antimicrobial Resistance and Use Surveillance System (WHO-GLASS) Report in 2021 states that out of 47 African countries, territories, and areas, only half (23/47, 49%) are enrolled in GLASS and only a third (15/47, 32%) reported information from the national surveillance system to GLASS [79]. The South East Asia region provides a better outlook: out of 11 countries, territories, and areas in South East Asia, all of them are enrolled in GLASS, and nine of them (81%) reported information from the national surveillance system [79]. However, some of the challenges to these programs include bias in sampling and data collection in these areas, which leads to gaps in knowledge about the AR situation at the global level.

Our findings have implications for clinical practice. Asking patients about their travel history in the past year might help clinicians in their decision-making process for choosing specific antibiotic protocols as the first-, second-, or third-line of treatment. Further, the use of a travel score, such as the one we have constructed, might be a straightforward way of quantifying the degree of risk due to travel. However, our travel score is still far from ready to be used in clinical practice in its current form. On the one hand, it does not include other details about the travel experience, such as reason for travel, length of stay, or place of residence within the visited location. It might be that individuals who travel abroad for business reasons are exposed to a very different set of environmental factors than those who travel to visit friends or family, partly because their consumption patterns might be different. Additionally, closer interactions with locals might increase the risk of direct or indirect exposure to AR bacteria such as ESBL-EC when sharing toilets with friends or family members, as opposed to staying at a hotel with private toilet facilities and frequent cleaning and disinfection.

According to our data, no other risk factor explored besides travels posed an effect on carriage of ESBL-EC. Antibiotics use is a risk factor for AR commonly mentioned in the literature [23,26]. We believe that one of the reasons why we were not able to estimate an effect for antibiotics use in our study is that, although these effects are relatively easy to identify in high-risk populations such as travelers, farmers, slaughterhouse workers, healthcare providers, or patients, the sample size needed to detect an effect in the general population would be considerably higher. Another potential reason is that the effect of antibiotics use on AR might not be detectable more than 6 months after travel. A recent study by Bunt et al. [66] in 4177 Dutch participants from the general population (four times the size of our study) showed a positive effect of antibiotics use for ESBL-EC carriage up to 6 months before study participation, but not at 6 to 12 months, nor more than a year before participation.

The main strength of our study is that, to our knowledge, this is the first international study across several countries that confirms travel risks for AR in the general population. Whereas many previously published studies have indeed reported travel as a risk factor for ESBL-EC carriage, our study was performed on a large sample stemming from the general population. These are generally healthy, working adults that were recruited without considering any specific high-risk factor for AR. Yet, we have found that travel is a risk factor for carriage of ESBL-EC, have characterized high-risk geographical areas for travels, and have estimated the magnitude of the effect of travelling to these areas. Additionally, although the study population was enrolled as part of the large trans-European cross-sectional AWARE study, it was assumed that individuals from the general population living more than 1000 m away from a local WWTP were not exposed to potential AR bacteria coming from such facilities. Therefore, we have a relatively large sample of participants drawn from the general population in Southern Germany, the Netherlands, and Romania. In contrast, other similar studies explored risk factors in large sample sizes from only one country [66], in specific high-risk populations, such as farmers [33,34,35,36,38,39,41,43,44] and slaughterhouse workers [32], healthcare workers and patients [40,45,46,48,49,50], or travelers [8,10,14,18,20,25,26,46], or in convenient samples of students [2,18,19,23]. Further, when exploring frequency of travel, we considered all areas of the globe, and did not limit ourselves to low-and-middle income countries or other areas that would have been otherwise considered a priori as high-risk areas for AR.

Some of the limitations of our study include a low response, especially in Germany and in the Netherlands, and a high proportion of missing values, especially in Germany, which lead to a relatively low statistical power for some potential risk factors and might limit the representativeness of our sample. We have used analytical tools, such as inverse probability of sampling weights, based on the response and multiple imputation to address these issues. Our potential risk factors were assessed by a questionnaire instead of by direct measurement or by cross-referencing with medical data, which might lead to recall bias and, thus, misclassification based on the risk factors. If this was the case, we would be erring on the conservative side by underestimating potential effects. Further, our sample might not be exempt from selection effects as our population was relatively young and highly educated. Age and socio-economic status (SES) might also play a role in our estimation of results from travel variables because we might assume that younger people travel more often and to different regions of the globe than older people, or because people of a higher SES might have the financial resources and freedom to travel more often than people of lower SES. In our study, we have included age and educational level (as a proxy for SES) in our regression models, thus adjusting for these potential confounders. 

## 5. Conclusions

In our study, we have identified travel to Northern Africa, Sub-Saharan Africa, Asia, and—to some extent—North America as independent risk factors for ESBL-EC carriage in a large sample of European individuals residing in Southern Germany, the Netherlands, and Romania. With our data, we were not able to identify other potential risk factors for carriage of ESBL-EC frequently mentioned in the literature such as the use of antibiotics within the past year, probably because the sample size needed to detect such effects in the general population would have to be at least about four times as large as ours. Further, we have developed a travel score that, although it needs refining to include information, such as reason for travel, length of stay, or place of residence, could be developed as a valuable tool in clinical practice when dealing with patients in need of an empirical treatment protocol with antibiotics. Questions about travel to Africa and Asia should continue to be routinely asked in clinical practice, as these travels are risk factors when considering antibiotic therapy.

## Figures and Tables

**Figure 1 ijerph-19-04758-f001:**
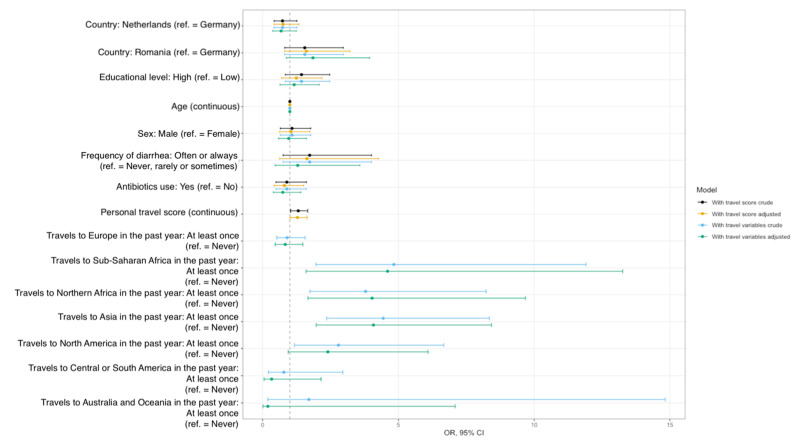
Risk factor analysis for carriage of ESBL-producing *E. coli* in stool samples.

**Figure 2 ijerph-19-04758-f002:**
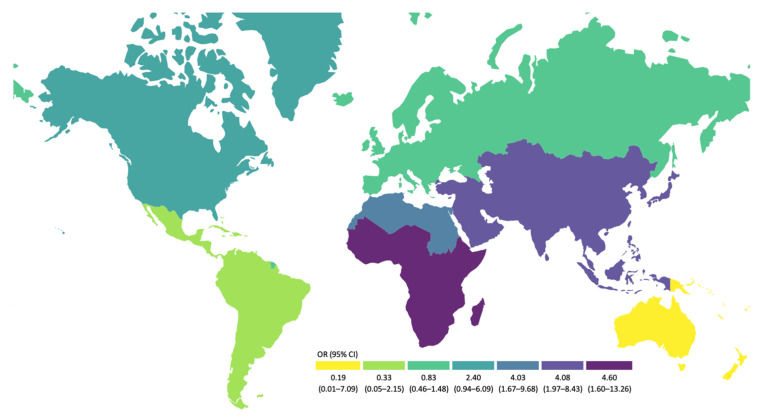
Travel areas as risk factors for ESBL-EC carriage (adjusted OR). Note: The European spot in South America corresponds to French Guiana.

**Table 1 ijerph-19-04758-t001:** Categorical descriptive characteristics of ESBL-producing *E. coli* carriers by country, n = 1183.

			Overall, n = 1074		Germany, n = 238		The Netherlands, n = 686		Romania, n = 150	
Missing Values for Stool Samples, n			109		95		3		11	
Variable	Missing	Level	ESBL_EC+, n (%)	*p*	ESBL_EC+, n (%)	*p*	ESBL_EC+, n (%)	*p*	ESBL_EC+, n (%)	*p*
ESBL-EC positives			81 (8)		20 (8)		42 (6)		19 (13)	
Sex	4	Female	47 (7)	0.814	12 (9)	1.000	25 (6)	0.871	19 (13)	1.000
		Male	34 (8)		8 (8)		17 (6)		9 (13)	
Highest educational level obtained ^a^	2	Low	19 (5)	**0.050**	6 (10)	0.602	13 (5)	0.196	0 (0)	0.217
		High	62 (9)		14 (8)		29 (7)		19 (14)	
Work with animals in the past year	35	No	75 (7)	0.752	18 (8)	1.000	40 (6)	0.659	17 (12)	0.555
		Yes	3 (8)		0 (0)		2 (8)		1 (17)	
Work at a farm in the past year	25	No	77 (7)	0.197	18 (8)	1.000	40 (6)	0.104	19 (13)	NA
		Yes	2 (18)		0 (0)		2 (22)		--- (---)	
Work at a slaughterhouse in the past year	20	No	79 (7)	1.000	18 (8)	NA	42 (6)	1.000	19 (13)	1.000
		Yes	0 (0)		--- (---)		0 (0)		0 (0)	
Work with manure in the past year	22	No	76 (7)	1.000	18 (8)	1.000	41 (6)	1.000	17 (12)	0.482
		Yes	2 (6)		0 (0)		1 (5)		1 (20)	
Patient contact or work with human tissues in the past year ^b^	20	No	52 (7)	1.000	12 (8)	1.000	29 (7)	0.738	11 (10)	0.133
		Yes	26 (7)		6 (7)		13 (6)		7 (21)	
Patient contact in the past year	20	Never	58 (7)	0.481	13 (8)	0.552	31 (6)	0.872	14 (12)	0.672
		Rarely or sometimes	11 (9)		3 (12)		5 (7)		3 (18)	
		Often or always	9 (6)		2 (5)		6 (5)		1 (12)	
Work with human tissues in the past year	16	Never	57 (7)	0.704	13 (8)	1.000	32 (6)	0.928	12 (10)	0.097
		Rarely or sometimes	13 (9)		3 (8)		6 (7)		4 (25)	
		Often or always	8 (7)		2 (7)		4 (5)		2 (20)	
Hospital visits as a patient in the past year	0	No	76 (8)	0.672	16 (8)	0.517	41 (6)	0.507	19 (13)	0.597
		Yes	5 (6)		4 (11)		1 (2)		0 (0)	
Hospital visits as a professional in the past year	0	No	78 (8)	0.761	19 (9)	1.000	41 (6)	1.000	18 (12)	0.336
		Yes	3 (8)		1 (5)		1 (7)		1 (33)	
Hospital visits as a visitor in the past year	0	No	79 (8)	0.690	18 (8)	0.169	42 (6)	1.000	19 (13)	1.000
		Yes	2 (9)		2 (22)		0 (0)		0 (0)	
Farm visits in the past year	4	No	71 (7)	0.578	17 (9)	0.773	35 (6)	0.097	19 (13)	0.596
		Yes	10 (9)		3 (7)		7 (11)		0 (0)	
Owning horses in the past year	139	No	77 (8)	0.722	20 (10)	0.605	40 (7)	1.000	17 (15)	1.000
		Yes	1 (4)		0 (0)		1 (7)		0 (0)	
Having dogs as pets in the past year	70	No	70 (9)	**0.011**	20 (11)	0.084	34 (7)	0.267	16 (16)	0.156
		Yes	9 (4)		0 (0)		7 (4)		2 (6)	
Having cats as pets in the past year	75	No	65 (9)	0.130	17 (10)	0.418	34 (7)	0.348	14 (15)	0.558
		Yes	13 (5)		3 (5)		7 (5)		3 (9)	
Use of antibiotics in the past year	0	No	60 (7)	0.685	13 (8)	1.000	36 (6)	0.544	11 (11)	0.289
		Yes	21 (8)		7 (8)		6 (5)		8 (17)	
Use of antacids in the past year	2	No	64 (8)	0.783	12 (7)	0.297	36 (7)	0.253	16 (13)	1.000
		Yes	17 (7)		8 (12)		6 (4)		3 (11)	
Surgeries in the past year	1	No	80 (8)	0.255	19 (9)	1.000	42 (6)	0.403	19 (13)	1.000
		Yes	1 (2)		1 (6)		0 (0)		0 (0)	
Self-reported frequency of diarrhea in the past year	4	Never, rarely or sometimes	74 (7)	0.223	17 (8)	0.069	38 (6)	0.347	19 (13)	1.000
		Often or always	7 (11)		3 (25)		4 (9)		0 (0)	
Self-reported health status in the past year	5	Good, very good or excellent	69 (7)	0.734	18 (8)	0.365	34 (6)	0.523	17 (13)	1.000
		Fair or poor	12 (8)		2 (13)		8 (7)		2 (11)	
Travel to high-risk areas for AR in the past year ^c^	8	No	36 (6)	0.012	6 (6)	0.336	17 (4)	**0.004**	13 (13)	0.791
		Yes	42 (10)		13 (10)		24 (10)		5 (10)	
Travels to Europe in the past year	5	Never	27 (9)	0.498	5 (12)	0.378	11 (6)	0.718	11 (13)	0.498
		Once	18 (8)		1 (2)		12 (7)		5 (18)	
		2–3 times	22 (6)		8 (8)		12 (5)		2 (8)	
		More than 3 times	12 (7)		5 (10)		7 (7)		0 (0)	
Travels to Bulgaria, Greece, Italy, or Slovenia in the past year	7	No	59 (8)	0.514	10 (8)	1.000	34 (6)	0.561	15 (15)	0.182
		Yes	19 (6)		9 (8)		7 (5)		3 (6)	
Travels to Sub-Saharan Africa in the past year	5	Never	73 (7)	**0.010**	19 (8)	1.000	36 (5)	**0.002**	18 (12)	NA
		Once	4 (19)		0 (0)		4 (22)		--- (---)	
		2–3 times	1 (33)		--- (---)		1 (33)		--- (---)	
		More than 3 times	1 (50)		--- (---)		1 (50)		--- (---)	
Travels to Northern Africa in the past year	6	Never	69 (7)	**0.001**	17 (7)	**0.013**	35 (5)	**0.019**	17 (12)	0.324
		Once	8 (20)		2 (25)		5 (17)		1 (33)	
		2–3 times	2 (50)		1 (100)		1 (33)		--- (---)	
		More than 3 times	0 (0)		--- (---)		0 (0)		--- (---)	
Travels to Asia in the past year	4	Never	63 (6)	**<0.001**	15 (7)	0.116	31 (5)	**<0.001**	17 (12)	0.408
		Once	13 (20)		3 (16)		9 (22)		1 (25)	
		2–3 times	2 (18)		1 (25)		1 (14)		--- (---)	
		More than 3 times	1 (50)		--- (---)		1 (50)		--- (---)	
Travels to North America in the past year	4	Never	73 (7)	**0.036**	17 (8)	**0.041**	38 (6)	0.146	18 (12)	NA
		Once	5 (17)		2 (20)		3 (16)		--- (---)	
		2–3 times	2 (25)		1 (50)		1 (17)		--- (---)	
		More than 3 times	0 (0)		--- (---)		0 (0)		--- (---)	
Travels to Central America or Mexico in the past year	6	Never	78 (7)	0.190	19 (8)	1.000	41 (6)	0.171	18 (12)	NA
		Once	0 (0)		0 (0)		0 (0)		--- (---)	
		2–3 times	0 (0)		0 (0)		--- (---)		--- (---)	
		More than 3 times	1 (50)		--- (---)		1 (50)		--- (---)	
Travels to South America in the past year	6	Never	77 (7)	0.149	18 (8)	0.287	41 (6)	0.126	18 (12)	NA
		Once	1 (6)		1 (25)		0 (0)		--- (---)	
		2–3 times	0 (0)		--- (---)		0 (0)		--- (---)	
		More than 3 times	1 (100)		--- (---)		1 (100)		--- (---)	
Travels to Australia or Oceania in the past year	6	Never	78 (7)	0.572	19 (8)	0.465	41 (6)	1.000	18 (12)	NA
		Once	1 (10)		1 (17)		0 (0)		--- (---)	
		2–3 times	0 (0)		0 (0)		--- (---)		--- (---)	
		More than 3 times	--- (---)		--- (---)		--- (---)		--- (---)	

Notes: ^a^ Educational level according to the International Standard Classification of Education (ISCED): Low = ISCED 0–2 (Pre-primary education to Lower secondary education), High = ISCED ≥ 3 (Upper secondary education to Doctoral or equivalent). ^b^ Work with human tissues in the past year: Includes self-reported contact with human tissues (e.g., blood, urine, sputum, feces, vomit, saliva, or primary cell lines). ^c^ Travels to high-risk areas for AR in the past year: Includes travels to North Africa, Sub-Saharan Africa, Asia, Central and South America, as well as the European countries Italy, Greece, Bulgaria and Slovenia. ESBL_EC+: Positive stool sample for Extended-Spectrum Beta-Lactamase-Producing *E. coli*; AR: Antibiotic Resistance. Bold highlighting means statistically significant at the *p* ≤ 0.05 level. Shown are the number of ESBL-EC carriers per variable and the percentage of ESBL-EC carriers relative to the total participants within the same level of that variable.

**Table 2 ijerph-19-04758-t002:** Numerical descriptive characteristics of ESBL-producing *E. coli* carriers by country.

		Overall, n = 1074			Germany, n = 238			The Netherlands, n = 686			Romania, n = 151		
Missing Values for Stool Samples, n		109			95			3			11		
Variable	Missings	ESBL_EC+	ESBL_EC−	*p*	ESBL_EC+	ESBL_EC−	*p*	ESBL_EC+	ESBL_EC−	*p*	ESBL_EC+	ESBL_EC−	*p*
n		81	993		20	218		42	644		19	131	
Age, years (median [IQR])	0	47 [34, 57]	51 [37, 60]	0.172	38 [31, 50]	49 [36, 58]	0.146	55 [42, 61]	54 [39, 61]	0.964	39 [34, 44]	40 [33, 50]	0.739
Travel score (mean ± SD, median [min, max]) ^a^	12	0.86 ± 1.60, 1 [0, 13]	0.46 ± 0.79, 0 [0, 15]	0.020	1 ± 0.94, 1 [0, 3]	0.64 ± 0.74, 1 [0, 6]	0.081	1.05 ± 2.07, 1 [0, 13]	0.42 ± 0.84, 0 [0, 15]	0.001	0.28 ± 0.46, 0 [0, 1]	0.38 ± 0.56, 0 [0, 3]	0.533

Notes: ^a^ Travel score was constructed based on frequency of personal travels to high risk areas for antibiotic resistance in the past year: Includes travels to North Africa, Sub-Saharan Africa, Asia, Central and South America, as well as the European countries Italy, Greece, Bulgaria, and Slovenia. The score is the sum of: zero points for not travelling to these areas in the past year, one point for travelling once to these areas in the past year, two points for travelling to these areas two or three times in the past year, and three points for travelling to these areas more than three times in the past year. Test used for bivariate hypothesis testing: Mann–Whitney test. ESBL_EC+: Positive stool sample for Extended-Spectrum Beta-Lactamase-Producing *E. coli*; ESBL_EC−: Negative stool sample for Extended-Spectrum Beta-Lactamase-Producing *E. coli*. IQR: Inter-quartile range. Bold highlighting means statistically significant at the *p* ≤ 0.05 level.

## Data Availability

The data presented in this study are available on request from the corresponding author. The data are not publicly available due to privacy reasons.

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
