# Peer review of "International Travel as a Risk Factor for Carriage of Extended-Spectrum β-Lactamase-Producing Escherichia coli in a Large Sample of European Individuals—The AWARE Study"

_ijerph, 2022, doi:10.3390/ijerph19084758_

Round 1
Reviewer 1 Report
General Comments:
Why wasn’t Sweden, a part of the AWARE program, included in this study? Why weren’t long term care facilities included since residency is a known risk factor?
Reference:
Tham J, Odenholt I, Walder M, Andersson L, Melander E. Risk factors for infections with extended-spectrum beta-lactamase-producing Escherichia coli in a county of Southern Sweden. Infect Drug Resist. 2013;6:93-97. Published 2013 Sep 19. doi:10.2147/IDR.S46290.
Specific comments:
Line 61 - The authors should illustrate the impact on patient care of having to use “last resort antibiotics.”
Lines 61-2 - What level (%) of increase has been seen worldwide? This figure should be included for comparison to Romania, Germany, and the Netherlands, if available.
Lines 166-67 - How many cultures were excluded that did not initially grow E. coli on either TBX or MacConkey plates?
Line 266 (Figure 1) - Font size of left figure labels needs to be increased to be readable.
Line 268 (Figure 2) - Font size of OR (95% CI) labels needs to be increased to be readable.
Lines 273-274 - The statement “Other studies in risk populations have found similar results.” should be referenced.
Line 278 - The statement “A 2017 prospective study done on 2001 Dutch travelers found out that 34.3%...” would read better as “A 2017 prospective study done on Dutch travelers (n = 2001) found out that 34.3%...”.
Line 331 - The authors should explain how sharing a toilet with family members increases the risk of acquiring ESBL-EC anymore that using a public toilet.
Line 346 - The authors claim that “The main strength of our study is that, to our knowledge, this is the first international 346 study across several countries that confirms travel risks for AR in the general population.” This may be true, but several previous studies have been published outlining international travel as an ESBL-EC risk factor.
Reviewer 2 Report
Dear Authors,
The presented study deals with very important issue of antimicrobial resistance spreading and factors behind this process. I found your research to be highly important and of high merit.
I have oly few minor remarks:
In the Abstract, please provide full names of all abbreviations used.
In the Introduction section:
59-61. This sentence is grammatically incorrect. Please rephrase
l. 69. How come nutrition is a risk factor for AR carriage? I suppose the Authors meant malnutrition or some specific type of nutrition? Also, health status and occupation – these are very unspecific terms. Please narrow it down, particularly in the case of occupation.
Please consider providing a graph showing the most important factors that you found in your study. Trying to comprehend large datasets presented in tables is both time consuming and some readers may find it difficult.
Reviewer 3 Report
1. Line 57-62: Requires reference.
2. The authors only performed phenotypic tests to detect ESBL E.coli, however, molecular methods should be employed to detect related genes (e. g blaCTX-M, blaNDM) in order to confirm the ESBL resistance traits.
3. In the confounding factor list, "completion of the course of antibiotic" should also be considered.
Round 2
Reviewer 1 Report
None.